# Current smoking alters phospholipid- and surfactant protein A levels in small airway lining fluid: An explorative study on exhaled breath

Emilia Viklund[1]*, Björn Bake[2], Laith Hussain-Alkhateeb[3], Hatice Koca Akdeva[1], Per Larsson[1], Anna-Carin Olin[1]

1 Occupational and Environmental Medicine, School of Public Health and Community Medicine, Institute of Medicine, Sahlgrenska Academy, University of Gothenburg, Gothenburg, Sweden, 2 Department of Respiratory Medicine and Allergology, Institute of Medicine, Sahlgrenska Academy, University of Gothenburg, Gothenburg, Sweden, 3 Global Health, School of Public Health and Community Medicine, Institute of Medicine, Sahlgrenska Academy, University of Gothenburg, Gothenburg, Sweden

* emilia.viklund@amm.gu.se

**Data Availability Statement:** All relevant data are within the manuscript and its Supporting Information files.

## Abstract

Small airways are difficult to access. Exhaled droplets, also referred to as particles, provide a sample of small airway lining fluid and may reflect inflammatory responses. We aimed to explore the effect of smoking on the composition and number of exhaled particles in a smoker-enriched study population. We collected and chemically analyzed exhaled particles from 102 subjects (29 never smokers, 36 former smokers and 37 current smokers) aged 39 to 83 years (median 63). A breathing maneuver maximized the number exhaled particles, which were quantified with a particle counter. The contents of surfactant protein A and albumin in exhaled particles was quantified with immunoassays and the contents of the phospholipids dipalmitoyl- and palmitoyl-oleoyl- phosphatidylcholine with mass spectrometry. Subjects also performed spirometry and nitrogen single breath washout. Associations between smoking status and the distribution of contents in exhaled particles and particle number concentration were tested with quantile regression, after adjusting for potential confounders. Current smokers, compared to never smokers, had higher number exhaled particles and more surfactant protein A in the particles. The magnitude of the effects of current smoking varied along the distribution of each PEx-variable. Among subjects with normal lung function, phospholipid levels were elevated in current smokers, in comparison to no effect of smoking on these lipids at abnormal lung function. Smoking increased exhaled number of particles and the contents of lipids and surfactant protein A in the particles. These findings might reflect early inflammatory responses to smoking in small airway lining fluid, also when lung function is within normal limits.

**Funding:** This research was supported by the Swedish Research Council for Health, Working Life and Welfare (A-CO, Grant nr:201600419, https://forte.se/en/), by the Heart and Lung foundation Sweden (A-CO, Grant nr: 20180209, https://www.hjart-lungfonden.se/), and by the Centre for Allergy Research Highlights Asthma Markers of Phenotype consortium (A-CO) which is funded by the Swedish Foundation for Strategic Research, the Karolinska Institute, AstraZeneca & Science for Life Laboratory Joint Research Collaboration, and the Vårdal Foundation. The funders had no role in study design, data collection and analysis, decision to publish, or preparation of the manuscript.

**Competing interests:** I have read the journal's policy and the authors of this manuscript have the following competing interests: The authors Emilia Viklund, Björn Bake, Per Larsson and Anna-Carin Olin are shareholders in PExA AB (www.PEXA.se). Anna-Carin Olin is one of the inventors of the PExA method and is a board member of PExA AB. This does not alter our adherence to PLOS ONE policies on sharing data and materials.

## Introduction

The patency of small airways (<2 mm internal diameter) is important for gas exchange. Pathological changes in this peripheral region give rise to few symptoms, thus, they are typically detected only in the later stages of lung disease. Small airways are largely inaccessible, and there is no easy method for retrieving a biological sample for the identification of biomarkers to use, for example, in screening. Biomarkers that reflect pathological changes in small airways may contribute to the early detection of adverse effects of airborne exposures, and possibly, to more effective treatment of lung diseases that affect small airways, such as chronic obstructive pulmonary disease (COPD). With this in mind, a method was developed for collection of particles in exhaled air (PEx).

Human breath aerosol contains endogenously generated droplets, also referred to as particles, of a very broad size distribution [1, 2]. PEx refers to exhaled particles in the size range of 0.5–5 μm that is being counted and collected, non-invasively, with the PExA method (i.e. the collection method) [3, 4]. The PEx content can be determined with various analytical assays [5–7].

In general, it is thought that a portion of small airways close upon expiration, and re-open upon inspiration. When small airways re-open during inspiration, the fluid lining the airways bursts, which generates particles that are small enough to be carried in the breath aerosol during the subsequent exhalation [4, 8]. The PExA method involves a specific breathing maneuver to maximize small airway closure, followed by small airway re-opening [4].

Airway closure increase with age and in lung diseases, like COPD [9]. The PEx number concentration, collected with the PExA method, is suggested to reflect the degree of airway closure followed by airway re-opening [4]. Particles larger than those collected with the PExA method are also exhaled, but these particles are possibly mainly generated in the upper airways [10].

Pulmonary surfactant, the major constituent of small airway lining fluid, is a complex mixture of phospholipids and proteins that prevents alveolar collapse by modulating surface tension. Moreover, the lining fluid is important in the host immune defense of the lungs; both as a barrier but also by specific proteins and lipids binding to inhaled material to enhance its elimination [11–13]. The two major phospholipids found in lung surfactant are the saturated di-palmitoyl-phosphatidyl-choline (DPPC), produced by alveolar type II cells, and the unsaturated palmitoyl-oleoyl-phosphatidylcholine (POPC).

Tobacco smoking is a well-known risk factor for COPD [14]; the smoke-associated inflammatory response is known to affect surfactant homeostasis by disturbing both lipid degradation and transport [15, 16]. Reduced DPPC and POPC contents have been found in the broncho-alveolar lavage (BAL) fluid collected from patients with COPD [17]. Taken together, these findings have raised interest in the interaction between tobacco exposure and changes in the lining fluid of small airways, especially at early stage, as there still is a lack of methods for early detection of disease, before damage to small airways is permanent.

Surfactant protein/lipid interactions are essential for maintaining surfactant homeostasis. The most abundant lung-specific protein in surfactant is surfactant protein A (SP-A), which has several important immunological functions. SP-A is mainly produced by the alveolar type II cells. SP-A promotes the re-uptake of oxidized surfactant lipids by alveolar type II cells [13] and is also an important opsonin of inhaled material [18]. Previous studies have suggested that SP-A might serve as a potential biomarker for early smoke-induced effects, even though somewhat contradictory results have been published [19–21]. A small study on former smokers with COPD showed that the SPA content in PEx deceased with worsening lung-function [22]. SP-A levels in PEx have also been shown to be highly correlated with that of BAL fluid [23].

Albumin is also found at relatively high concentrations in small airways. A previous study showed that content of albumin in PEx was decreased in those subjects with asthma that also

had small airway dysfunction [24]. Therefore, albumin may be of interest when studying the effect of tobacco smoke on small airway lining fluid.

Tobacco smoke is a mixture of thousands of toxic compounds, and the composition of small airway lining fluid is complex. There is likely to be an intricate interaction between respiratory irritants and small airway function. In addition, the influences of other factors on small airway lining fluid, such as age and lung function, increase the complexity. Based on this and on previous studies, we hypothesized that the effect of smoking varies across the distributions of surfactant lipids and proteins in small airway lining fluid and that the effects are associated with lung function.

The present study aimed to explore potential biomarkers for inflammation in small airway lining fluid; i.e. effects on the composition of lipids (DPPC and POPC) and proteins (SP-A and albumin) in PEx and on the PEx number concentration, in cigarette smokers. Furthermore, we wanted to examine if the novel PExA method could identify alterations, associated with tobacco smoking, before lung function is affected.

## Materials and methods

### Study population

102 subjects, aged 39–83 years, were recruited to participate in an extended study protocol in the follow-up study of the population-based INTERGENE-ADONIX cohort [25]. Recruited subjects had all given their written informed consent in the follow-up study to be contacted for participation in the extended study protocol. Eligible for inclusion were subjects whom had managed to perform spirometry and had reported their smoking status and smoking history in the follow-up study. Exclusion criteria were ongoing respiratory tract infection, myocardial infarction past month and/or pregnancy in last trimester. All current smokers were invited to participate, whereas former- and never-smokers were invited randomly. The subjects were classified as current-, former- and never smokers based on smoking history; current smokers reported to have smoked cigarettes on a regular daily basis for at least one year at the time of the clinical examination; former smokers had smoked on a regular basis, but had stopped smoking for one year or more before the clinical examination. Never smokers had never smoked on a regular basis.

The regional ethics board at Gothenburg University approved the study protocol (application number 626–13). Written informed consent was obtained from all participating subjects. The clinical characteristics of the study population, according to smoking status, are presented in Table 1.

### Study design

This cross-sectional study was conducted between June 2014 and September 2016 in Gothenburg, Sweden. The clinical examination comprised an exhaled air analysis, blood sample, lung function tests, and the completion of questionnaires. Current smokers were instructed to refrain from smoking 1 h prior to the clinical tests. All participants were instructed to withhold from taking long-acting β2-bronchodilators and inhaled gluco-corticoids for at least 24 h and short-acting β2-broncho-dilators for at least 12 h prior to the clinical tests. Included participants had no respiratory tract infections within three weeks prior to the clinical examination.

### Measurements

**FeNO.** The Fraction of Exhaled Nitric Oxide (FENO) was measured at an expiratory flow of 50 mL/s with a chemiluminescence analyzer (NIOX VERO, Aerocrine AB, Stockholm,

**Table 1. General and clinical characteristics of study population, subdivided according to smoking category.**

| | All | Never smokers | Former smokers | Current smokers |
|---|---|---|---|---|
| N | 102 | 29 | 36 | 37 |
| Sex females/males [n] | 49/53 | 14/15 | 15/21 | 20/17 |
| Age [years] | 62.5 (51.8–69.0) | 52.0 (47.0–66.5) | 69.5 (61.3–72.0) | 59.0 (53.0–66.5) |
| BMI [kg/m$^2$] | 25.3 (23.6–27.8) | 25.3 (23.2–26.9) | 26.7 (24.4–30.0) | 24.7 (22.6–27.5) |
| Smoking history [pack years] | 19.0 (0–33.0) | 0 | 23.5 (15.3–35.3) | 28.0 (18.0–40.0) |
| FENO50 [ppb] | 18.0 (13.0–23.0) | 19.0 (16.5–25.5) | 19.0 (13.0–25.0) | 16.0 (11.0–21.0) |
| CRP [mg/L] | 1.3 (0.8–2.6) | 1.0 (0.5–1.8) | 1.3 (0.7–2.6) | 1.7 (1.1–3.1) |
| $N_2$-slope [%N2/L] | 1.5 (1.1–3.5) | 1.2 (0.8–1.5) | 2.5 (1.4–4.4) | 1.9 (1.2–3.5) |
| FEV1 [z-score] | -1.3 (-2.0- -0.4) | -0.8 (-1.7- -0.2) | -1.5 (-2.1- -0.8) | -1.2 (-2.1- -0.1) |
| FVC [z-score] | -0.5 (-1.4–0.1) | -0.1 (-1.3–0.3) | -1.2 (-1.5–0.0) | -0.6 (-1.4–0.3) |
| FEV1/FVC post BD [%] | 0.73 (0.68–0.79) | 0.78 (0.69–0.80) | 0.73 (0.66–0.78) | 0.73 (0.69–0.78) |

Values are presented as median with inter quartile range, if nothing else stated. FENO50; Fraction of Exhaled Nitric Oxide at an expiratory flow of 50 mL s-1, CRP; C-Reactive Protein, $N_2$-slope; slope III of single breath washout, FEV1; Forced Expiratory Volume in the 1st second, FVC; Forced Vital Capacity, FEV1; Forced Expiratory Volume in the 1st second, FVC; Forced Vital Capacity.

Sweden). Measurements were in accordance with American Thoracic Society (ATS) and European Respiratory Society (ERS) recommendations [26].

**Spirometry.** Spirometry was performed with a Spirare device (SPS3110 sensor and Spirare 3 software; Diagnostica AS, Oslo, Norway), before and after bronchodilation with 1.5 mcg terbutaline (Bricanyl; AstraZeneca; Sweden), in accordance with the ATS/ERS criteria [27]. The forced expired volume in the first second ($FEV_1$) and the forced expiratory vital capacity (FVC) were measured, and the ratio $FEV_1$/FVC was calculated. Predicted normal values were based on local reference values from Brisman et al [28, 29], and the results are expressed in terms of percent predicted (% pred), the z-score (z) or the Lower Limit of Normality (LLN).

**Nitrogen single breath washout.** Nitrogen single breath washout ($N_2$SBW) was performed before bronchodilation, with an Exhalyzer D (Eco Medics AG). The target was three technically acceptable trials. Each trial included a full expiration to residual volume (RV), followed by a slow inspiration (maximum 500 mL/s) of 100% oxygen to TLC, and finally, a slow exhalation (maximum 500 mL/s from TLC to RV. A linear regression of data between 25–75% of the exhaled volume was performed to obtain the alveolar nitrogen slope ($N_2$-slope). Trials were acceptable when the coefficient of variation in the expiratory vital capacity was less than 10% between trials. The mean value of the accepted trials was used in the analyses.

**Exhaled particles.** PEx collections were performed after bronchodilation, with the PExA instrument (PExA AB, Göteborg, Sweden), according to the method described in detail previously [3]. The PExA instrument does not collect all particle sizes found in the breath. In this study, particles in a size interval of 0.5–5.0 μm were collected and referred to as PEx. In brief, the PExA instrument included an optical particle counter (Grimm 1.108, Grimm Aerosol Technik GmbH, Ainring, Germany) and a collection plate covered with a thin membrane of hydrophilic polytetrafluoroethylene (PTFE) (FHLC02500, Millipore, Billerica, MA, USA), in a modified multi-stage impactor (PM10 Impactor, Dekati Ltd., Tampere, Finland). With a diverter valve, the operator can divert the flow of the exhaled breath, either into the PExA instrument for sampling or back to ambient air, when sampling only selected exhalations. Subjects wore a nose clip and repeatedly performed a standardized breathing maneuver. The breathing maneuver started with an exhalation to residual volume (RV), a breath-hold for 5 s, a rapid inhalation to total lung capacity (TLC), and then, this was immediately followed by a deep

exhalation at a spontaneous flow rate. PEx was sampled only during this final exhalation of the breathing maneuver. Between breathing maneuvers, subjects breathed tidally in filtered, particle-free air. The collection continued until 120 ng of PEx mass was obtained [30]. The PTFE membrane with sampled PEx was divided into two halves, which were transferred to separate, 2 mL polypropylene cryotubes (Sarsteds, Nümbrecht, Germany) and stored at -80°C, prior to chemical analysis. The PEx number concentrations are expressed as n $^*$1000 (kn) per litre of exhaled breath, kn/L, and are referred to as number PEx.

**Chemical analyses of exhaled particles.** Chemical analyses of the proteins in PEx (SP-A and albumin) were performed with enzyme- linked immunosorbent assays (ELISAs), according to the protocol described by Kokelj and colleagues [31]. In brief, PEx were extracted from the PTFE membrane samples by adding extraction buffer. Levels of SP-A and albumin in extracted PEx samples were determined with a human SP-A ELISA kit (Lot nr; E-15-108, Product nr; RD191139200R, BioVendor, Brno, Czech Republic) and a human albumin ELISA kit (Lot nr; 19, Part number; E-80AL,Immunology Consultant Laboratory, Newberg, OR, USA), according to the manufacturer´s instructions, with minor modifications.

Chemical analyses of the phospholipids in PEx (DPPC and POPC) were performed with a triple quadrupole mass spectrometer (Sciex API3000, AB Sciex, Canada), equipped with an electrospray ion source operating in positive mode, as described in detail previously [7]. Briefly, internal standards were added to each sample before extraction. Extracted samples were introduced to the ion source with a flow gradient method, but without chromatography separation. DPPC and POPC were quantified based on a calibration curve with a linear regression model. In each run PTFE substrates spiked with 25 pico mol DPPC and POPC standards (an amount representative of the study samples) were analyzed to monitor method performance. Based on 22 measurements distributed throughout the study the average recovery was 81% and 104% for DPPC and POPC respectively. The reproducibility RSD% was 6.4 and 8.3 for DPPC and POPC respectively.

The concentrations of SP-A, albumin, DPPC, and POPC in PEx were calculated as the weight-percent of PEx (wt%), by dividing the mass of the analyte by the PEx mass. Number PEx and the concentrations of DPPC, POPC, SP-A, and albumin in PEx are all referred to as PEx variables.

## Statistical methods

Statistical analyses were performed with IBM SPSS 26.0 software (SPSS, Chicago, IL). Number PEx were not normally distributed and therefore non-parametric tests were used to analyze both composition of PEx and number PEx. The Kruskal-Wallis test was used for testing differences between smoking categories. The Spearman rank correlation coefficient was employed to test linear relationships. Quantile regression was performed to assess associations between smoking status and different segments of the distribution of each PEx variable (i.e. number PEx and the composition of DPPC, POPC, SP-A, and albumin in PEx). Estimates of coefficient with confidence intervals in quantile regression indicate the change in the value at a modeled quantile of the dependent variable (e.g. number PEx) for each unit change in the independent variable (e.g. current smoker compared to never smoker). Age and sex were considered to have biological relevance with the outcomes and predictors, therefore; these factors were used as confounders in the quantile regression model. The effect of smoking status on the distribution of each PEx variable was considered significant when the confidence interval was separated from zero on the axis of the estimate. Each PEx variable was analyzed in independent models.

The association between smoking and the different PEx variables were also analysed in models stratified for lung function. Lung function was defined as normal when values of

FEV1, FVC, and FEV1/FVC were ≥LLN. Abnormal lung function was defined as FEV1, FVC, and/or FEV1/FVC values <LLN. These sub-analyses were performed independently of each other. A p-value < 0.05 was regarded statistically significant.

## Results

In crude regression analysis, current smokers exhaled significantly higher concentration of number PEx and had higher concentration of DPPC and SP-A in PEx, compared to never smokers (Table 2). Current smokers also exhaled higher concentration of number PEx and had a higher concentration of POPC in PEx compared to former smokers. No significant differences in any of the PEx variables were found between never-smokers and former smokers.

In multiple regression analysis, adjusted for age and sex, the associations between smoking status and each PEx variable were estimated along the entire distribution of the different PEx variables (Fig 1). In current smokers, in comparison to never smokers, number PEx was increased, and the magnitude of the effect was amplified with increasing number PEx, as illustrated by the change in estimates between quantile 0.5 and 0.75 (Coef 11.1 kn/L, CI: 3.3–18.9 and Coef 19.3 kn/L, CI: 4.5–34.1, respectively) (Fig 1A). SP-A in PEx was also found to be higher in current smokers but only in the upper half of the quantiles, and the magnitude of the effect increased with increasing SP-A, as illustrated by the change in estimates between quantile 0.5 and 0.75 (Coef 0.9 wt%, CI: 0.1–1.6 and Coef 1.1 wt%, CI: 0.4–1.7, respectively) (Fig 1B). The estimates with confidence interval along the distribution of DPPC and POPC concentrations in PEx showed similar patterns for current and former smokers; however, only just about significantly increased in current smokers and only in restricted parts along the distributions. For instance, DPPC was significantly increased in quantile 0.25 (Coef 2.0 wt%, CI: 0.4–3.6), whereas POPC was significantly increased in quantile 0.5 (Coef 0.9 wt%, CI: 0.3–1.4), in current compared to never smokers (Fig 1D and 1E). No associations were found between albumin in PEx and current or former smokers, in comparison to never-smokers (Fig 1C). In parallel, no associations were found regarding former smokers and any of the PEx variables.

Correlations between the different PEx variables and subject characteristics were evaluated in current smokers (Table 3). An increase in SP-A in current smokers was correlated with a decrease in the FEV1/FVC ratio. None of the PEx variables were correlated with either the $N_2$-slope (%N2/L) or the smoking history (pack years).

### Associations between PEx-variables and lung function

Among subjects with normal lung function (FEV1, FVC, and FEV1/FVC ≥LLN), no significant associations were found between number PEx and current smokers, in comparison to

**Table 2. PEx-variables in study population, subdivided according to smoking category.**

|  | Never smokers | Former smokers | Current smokers | p-value |
|---|---|---|---|---|
| **number PEx [kn/L]** | 13.2 (7.8–20.5) | 14.1 (10.7–20.8) | 20.8 (12.4–35.7)[a] | 0.011 |
| **DPPC [wt %]** | 10.3 (8.5–11.7) | 10.6 (8.6–11.6) | 11.3 (10.2–12.9)[a] | 0.025 |
| **POPC [wt %]** | 2.9 (2.5–4.0) | 3.1 (2.4–3.6) | 3.7 (3.2–4.2)[a,b] | 0.008 |
| **SP-A [wt %]** | 3.1 (2.4–3.5) | 3.2 (2.4–3.9) | 3.9 (2.6–4.4)[a] | 0.037 |
| **Alb [wt %]** | 7.5 (5.4–8.8) | 8.8 (6.4–10.3) | 6.9 (5.4–10.0) | 0.184 |

Data are presented as median with interquartile range (Q1-Q3). PEx: Particles in Exhaled Air; kn/L: thousand number exhaled PEx per liter exhaled air; DPPC: Dipalmiotoylphosphatidylcholine; wt%: weight percent of PEx; POPC: Palmitoyl-oleoylphosphatidylcholine; SP-A: Surfactant Protein A; Alb: Albumin. P-values based on Kruskal-Wallis test followed by Bonferronis multiple comparisons tests of significant difference (p<0.05)between

[a]current smokers and never smokers

[b]current smokers and former smokers.

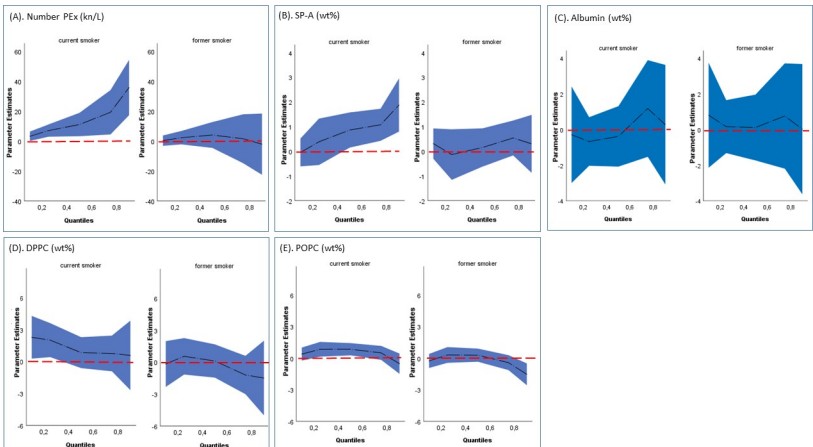

**Fig 1. Levels of number PEx (kn/L), phospholipids and proteins in PEx (wt%) among current- and former smokers compared to never-smokers.** Plots illustrating age- and sex-adjusted quantile regression estimates of current- and former smokers to never-smokers (the red dotted line). (A) number PEx (kn/L), (B) SP-A (wt%), (C) Albumin (wt%), (D) DPPC and (E) POPC (wt%) were analyzed separately. Quantiles on x-axis refers to the distribution of the PEx-variable studied. Estimates from quantile regression denoted by black dotted line with blue confidence intervals.

that of never-smokers, in multiple regression analysis adjusted for age and sex (Fig 2A, upper graph). However, when restricting the analysis to subjects with abnormal lung function (FEV1, FVC, and/or FEV1/FVC <LLN), current smokers had markedly higher number PEx than never- smokers (Fig 2A, lower graph). The magnitude of the effect was amplified with increasing number PEx, as illustrated by the different estimates in quantile 0.5 compared to quantile 0.75 (Coef 14.2 kn/L, CI: 4.4–24.0 and Coef 33.4 kn/L, CI: 15.9–50.9, respectively). Associations between smoking status and distribution of SP-A in these restricted analyses, showed similar patterns as in the entire study population, although only barely significantly increased in a small window of the higher quantiles of the SP-A distribution (Fig 2B). Current smokers with normal lung function had higher levels of DPPC than never-smokers along its entire distribution (Fig 2D, upper graph), and POPC levels, but with a large and not significant confidence interval in the upper quantiles (Fig 2E, upper graph). No significant associations were found regarding current smokers and either of the lipids, in the analysis restricted to subjects with abnormal lung function (Fig 2D and 2E, lower graphs). Albumin showed no effect of current smoking (Fig 2C).

**Table 3. Correlation coefficients (spearman) between each PEx parameter and characteristics in current smokers (n = 37).**

| Variables | number PEx [kn/L] | DPPC (wt%) | POPC (wt%) | SP-A (wt%) | Alb (wt%) |
|---|---|---|---|---|---|
| **Age [years]** | -0,274 | 0,131 | 0,088 | 0,260 | 0,362* |
| **smoking history [packyears]** | -0,047 | 0,027 | -0,058 | 0,145 | 0,062 |
| **FENO50 [ppb]** | -0,336 | 0,045 | 0,047 | 0,077 | -0,033 |
| **CRP [mg/L]** | 0,160 | -0,357* | -0,321 | 0,054 | -0,157 |
| **$N_2$-slope [%$N_2$/L]** | -0,028 | 0,206 | 0,032 | 0,032 | -0,091 |
| **FEV1 [z-score]** | -0,034 | -0,081 | -0,077 | -0,101 | 0,071 |
| **FVC [z-score]** | -0,154 | -0,140 | -0,156 | 0,098 | 0,165 |
| **$FEV_1$/FVC post BD** | 0,188 | -0,020 | 0,119 | -,411* | -0,282 |

Data expressed as spearman r value.

* p< 0.05.

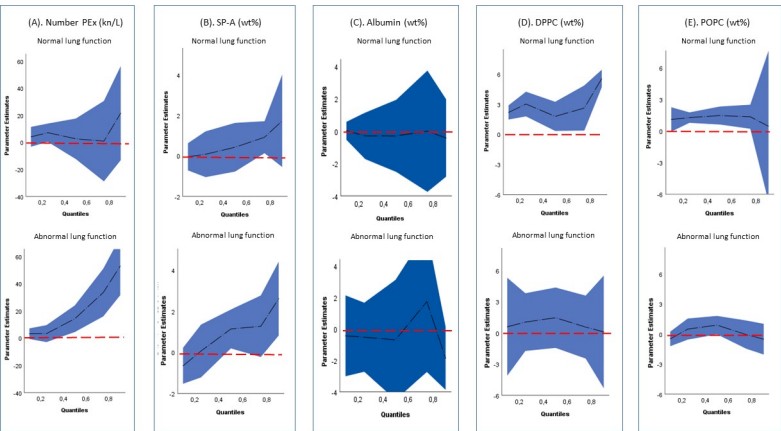

**Fig 2. PEx-variables among current- compared to never-smokers at normal and abnormal lung function.** Plots illustrating age- and sex-adjusted quantile regression estimates of current smokers to never smokers (the red dotted line), stratified on normal and abnormal lung function (n = 35 and n = 31, respectively). (A) number PEx (kn/L), (B) SP-A (wt%), (C) Albumin (wt%), (D) POPC (wt%) and (E) POPC (wt%) were analyzed separately, as were the sub-groups. Quantiles on x-axis refers to the distribution of the PEx-variable studied. Estimates from current smokers compared to never smokers presented at normal lung function (n = 19 and n = 16, respectively) and abnormal lung function (n = 18 and n = 13, respectively). Estimates from quantile regression denoted by black dotted line with blue confidence intervals.

## Discussion

The effects of long-term smoking on the composition of the lining fluid of small airways, like-wise the number of formed droplets of this fluid (i.e. PEx; particles in exhaled air), have been studied using the PExA method. Current smokers with normal lung function had increased concentrations of DPPC and POPC in PEx, in comparison to never smokers. Also, smokers tended to have higher levels of SP-A in PEx, irrespective of lung function. The number of PEx was markedly increased in current smokers compared to never smokers, and most apparent among subjects with abnormal lung function. The magnitude of these associations varied over the distribution of the different PEx variables.

We used quantile regression to study smoking and its association with the distribution of each PEx variable as the associations were suspected not to be equal along the different quantiles. It is well known that the effect of smoking differ between subjects, and in this cross-sectional setting, we assumed that the inflammatory responses and remodeling processes as a result of smoking, were unequal distributed among the included subjects. It seems likely that smoking may induce changes in the surfactant composition in small airways as a protective mechanism, but at certain stage this will not suffice in susceptible individuals, and pathological processes will take over. By this time the surfactant function will deteriorate, airways start to close, emphysema will develop and the lung function will start to decline. This was indicated by the present results, with a significant association between current smoking and higher DPPC in PEx among subjects with normal lung function as compared to no association between smoking and lipids in PEx in subjects with abnormal lung function (Fig 2D). Previous finding by Laerstad et al [22], that subjects with more severe COPD (GOLD II-IV) had lower number of exhaled particles, further support this.

Consistent with these effects on the lipid homeostasis after smoking, previous experiments on smoke-exposed mice have shown that smoke disturbed the capacity of alveolar macrophages to take up oxidized lipids [15]. However, longitudinal studies are needed to elucidate these associations in humans, and the present findings only allows one to speculate about the

effect of the irritants in tobacco smoke on the synthesis, secretion, and/or reuptake of surfactant lipids.

To date, knowledge is limited on changes in the small airways lining fluid in humans. In COPD, Agudelo and colleagues [17] recently showed a correlation between lung function and a reduction in surfactant lipids, based on the BAL fluids of former smokers. In present study, DPPC and POPC in PEx were found increased in current smokers throughout almost the entire distribution of these lipids, when restricting the analyses to subjects with normal lung function. Recent findings by Hussain-Alkhateeb et al [32], using quantile regression (50th percentile), also showed increased content of DPPC and POPC in PEx in current smokers (n = 17). Important to note, normal spirometry does not exclude subjects with affected lungs. A previous study on smokers with preserved spirometry (n = 4388) showed half of the smokers to have radiologic abnormalities, from which a large portion had evidence of emphysema or airway wall thickening [33]. This highlights the need for novel methods to easily detect changes in lung function.

The SP-A distribution shifted towards higher values in current smokers, compared to never smokers, and the association was stronger in the higher quantiles (Fig 1B). In a previous small exploratory study in former smokers with COPD, SP-A in PEx decreased significantly with worsening FEV1 [22]. In present study, SP-A correlated negatively to the FEV1/FVC ratio in current smokers (Table 3). In subjects with normal lung function, there was however no significant association between SP-A and smoking (Fig 2B, right hand graph), although the estimates were in the same direction as in subjects with abnormal lung function. SP-A seems thus to be associated with lung function, but also with on-going exposure.

SP-A is known to be important in enhancing the phagocytosis of inhaled toxins, and thus, it facilitates toxin clearance, for example, by alveolar macrophages. Accordingly, we speculated that high levels of SP-A in the surfactant of small airways lining fluid, here reflected as high content of SP-A in PEx, was a response to the inhalation of respiratory toxins from tobacco smoke. Thus, as high SP-A levels would speed up the removal of these toxins, it may be a favorable response.

The abundance of albumin in PEx was not affected by smoking status. One might assume it should increase with smoking, due to the increased leakage from the systemic circulation in smokers, but this did not seem to be the case. Our finding was supported by results from an earlier study by Schmekel et al [34].

The number PEx per liter of exhaled air was substantially increased in current smokers compared to never smokers (Fig 1A). This finding suggested that current smoking might increase the number of small airways that close and re-open. This hypothesis is consistent with previous findings, reporting increased closing volumes in smokers [35, 36]. The closing volume is the lung volume during an expiration, when a substantial number of airways close. It is highly likely that the larger the closing volume, the larger the number of airways that close and re-open. Thus, the high number PEx that we found in smokers might be explained by the extent of airway closure. The primary cause of increased closing volume in smokers has been attributed to smoking-induced loss of lung elastic recoil [37], which facilitates the closing of small airways.

Another possible mechanism underlying the increased number PEx in smokers might be the effects of smoking on the physical properties of surfactant *per se*. In computational studies on liquid film burst, an increased surface tension resulted in higher concentrations of droplets [38, 39]. Thus, if tobacco smoke increases the surface tension in surfactant, the number PEx might increase in smokers. Potentially, an increased number of exhaled particles might be due to both the increased number of airways that close and re-open and the altered surfactant homeostasis.

In analysis restricted to subjects with normal lung function, no effect of smoking on number PEx was shown. However, in the analysis on subjects with abnormal lung function, there was a strong association between current smokers and an increase in number PEx, indicating that small airways close and re-open easier in current smokers with abnormal lung function.

Intra-pulmonary ventilation inhomogeneity, assessed with the $N_2$-slope (%N2/L), was not correlated with any PEx variables in current smokers (Table 3). In previous studies, the $N_2$-slope were shown to correlate with pathological changes in the small airways in smokers [40, 41]. However, the $N_2$-slope reflects structural changes that not necessarily overlap with inflammatory changes in small airways; potentially, this lack of overlap might explain the lack of correlation found in the present study.

Taken together, the increased number PEx and the altered levels of DPPC, POPC and SP-A in the PEx of current smokers may reflect a disturbed homeostasis of small airway lining fluid caused by smoking. Protein-lipid interactions are essential for maintaining the homeostasis of small airways lining fluid. For instance, SP-A promotes the re-uptake of inactivated surfactant lipids by alveolar type II cells [13]. The magnitudes of the effects of smoking varied along the distributions of each PEx variable, and we speculated that this variation might have reflected different stages of inflammatory response in the small airways lining fluid. These associations need however to be analyzed more in depth in a larger material with longitudinal design, before any stronger conclusions can be drawn.

Some strengths of the present study are worth pointing out. First, we measured PEx instead of BAL fluid. The PEx matrix consists of undiluted lining fluid from small airways that, due to the non-invasive methodology, provide a strong advantage over sampling and measuring potential biomarkers in BAL fluid, which contains diluted concentrations. Additionally, we expressed the concentrations of the different biomarkers in PEx in terms of the weight percent of the sample, thus, corrected for differences in the mass of the sample. Finally, to avoid systematic bias in present study, all samples were handled in the same way, but they were analyzed in a randomized order.

The present study also have limitations, the main being the small number of included subjects limiting the generalizability of the results. Especially the number of current smokers with abnormal lung function was low, which limited the analysis. The intra- and inter-individual variations in PEx-variables, collected with the standardized breathing maneuver, has not been addressed in current study. For practical reasons, it was not possible to perform repeated measures or measures at the same time-point during the day, which might have reduced the bias of diurnal variations in PEx variables if taken into account [31]. However, the recently published study by Hussain-Alkhateeb et al showed similar results as ours [32].

## Conclusions

Tobacco smoking seems to influence on the composition of the lining fluid of small airways, and both the phospholipids DPPC and POPC as well as SP-A were increased in current smokers. The magnitude of these effects varied along the distribution of the PEx variables and seemed to be associated with lung function. The differences we observed between smokers and never smokers are likely to be associated with small airway inflammation, which is known to be induced by smoking, but further analyses are needed to explore these associations more in depth. As the PExA method is non-invasive and easy to apply, it may help us to identify novel biomarkers for disease at an early stage. At individual level, it is easy to neglect that smoking implies an increased risk for severe outcomes, but if there are signs of ongoing inflammation and an improved individual risk-assessment, it may help people to quit smoking. In a longer run, PExA might be a useful tool for the screening of large populations and may facilitate the

identification of subjects at risk of developing severe diseases affecting small airways, such as COPD.

## Supporting information

**S1 File. Data set.**
(PDF)

## Acknowledgments

The authors are grateful to all subjects for their participation in the present study. The authors also gratefully acknowledge the work of Helen Friberg, Lillvor Ivarsson Scherman, Annica Claesson and Marianne Andersson for examining all study participants, and, Jeong-Lim Kim for organizing all the collected data.

## Author Contributions

**Conceptualization:** Emilia Viklund, Björn Bake, Anna-Carin Olin.

**Data curation:** Emilia Viklund, Björn Bake, Laith Hussain-Alkhateeb, Hatice Koca Akdeva, Per Larsson.

**Formal analysis:** Emilia Viklund, Björn Bake, Laith Hussain-Alkhateeb, Hatice Koca Akdeva, Per Larsson.

**Funding acquisition:** Anna-Carin Olin.

**Investigation:** Emilia Viklund, Björn Bake, Hatice Koca Akdeva, Per Larsson.

**Methodology:** Emilia Viklund, Björn Bake, Laith Hussain-Alkhateeb, Hatice Koca Akdeva, Per Larsson.

**Project administration:** Anna-Carin Olin.

**Resources:** Anna-Carin Olin.

**Software:** Laith Hussain-Alkhateeb.

**Supervision:** Björn Bake, Anna-Carin Olin.

**Validation:** Emilia Viklund, Björn Bake, Laith Hussain-Alkhateeb, Hatice Koca Akdeva, Per Larsson, Anna-Carin Olin.

**Visualization:** Emilia Viklund.

**Writing – original draft:** Emilia Viklund, Björn Bake, Anna-Carin Olin.

**Writing – review & editing:** Emilia Viklund, Björn Bake, Laith Hussain-Alkhateeb, Hatice Koca Akdeva, Per Larsson, Anna-Carin Olin.

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
