## [Decision Letter · Decision Letter 0]

12 Apr 2021

PONE-D-21-07178

Current smoking alter phospholipid- and surfactant protein A levels in small airway lining fluid: an explorative study on exhaled breath

PLOS ONE

Dear Dr. Viklund,

Thank you for submitting your manuscript to PLOS ONE. After careful consideration, we feel that it has merit but does not fully meet PLOS ONE’s publication criteria as it currently stands. Therefore, we invite you to submit a revised version of the manuscript that addresses the points raised during the review process. Both reviewers raised some signifcant concenrs which are mainly attibuted to methodological issues as well as to the way that results were interpreted. Please also revse the statistica analysis.

We look forward to receiving your revised manuscript.

Kind regards,

Stelios Loukides

Academic Editor

PLOS ONE

Journal Requirements:

2. In your Methods section, please provide additional information about the participant recruitment method and the demographic details of your participants. Please ensure you have provided sufficient details to replicate the analyses such as: a) the recruitment date range (month and year), b) a description of any inclusion/exclusion criteria that were applied to participant recruitment, d) a description of how participants were recruited, and e) descriptions of where participants were recruited and where the research took place.

3. Please provide a sample size and power calculation in the Methods, or discuss the reasons for not performing one before study initiation.

4. Please provide the product number and any lot numbers of the ELISA kits purchased for your study.

5. Please note that PLOS does not permit references to 'data not shown.' Authors should provide the relevant data within the manuscript, the Supporting Information files, or in a public repository. If the data are not a core part of the research study being presented, we ask that authors remove any references to these data.

Please add a citation to support this phrase or upload the data that corresponds with these findings to a stable repository (such as Figshare or Dryad) and provide and URLs, DOIs, or accession numbers that may be used to access these data.

6. During your revisions, please confirm whether the wording in the title is correct and update it in the manuscript file and online submission information if needed. Specifically, please consider whether it should read "Current smoking alters..." rather than "alter".

Reviewers' comments:

Reviewer's Responses to Questions

**Comments to the Author**

1. Is the manuscript technically sound, and do the data support the conclusions?

Reviewer #1: No

Reviewer #2: Yes

2. Has the statistical analysis been performed appropriately and rigorously? 

Reviewer #1: No

Reviewer #2: Yes

3. Have the authors made all data underlying the findings in their manuscript fully available?

Reviewer #1: Yes

Reviewer #2: Yes

4. Is the manuscript presented in an intelligible fashion and written in standard English?

Reviewer #1: No

Reviewer #2: Yes

5. Review Comments to the Author

Reviewer #1: Regarding the submitted manuscript Current smoking alter phospholipid- and surfactant protein A levels in small airway lining fluid: an explorative study on exhaled breath-

The authors aimed to explore the effect of smoking on the composition and number of exhaled particles in a smoker-enriched study population.

The study consists of 102 subjects (29 never smokers, 36 former smokers and 37 current smokers) aged 39 to 83 years (median 63). Samples were optioned at one time point for all included patients.

The contents of surfactant protein A and albumin in exhaled particles was quantified with immunoassays and the contents of the phospholipids dipalmitoyl- and palmitoyl-oleoyl- phosphatidylcholine with mass spectrometry.

I have some major concern regarding the current manuscript:

1. Please explain the rational for analyzing number of exhaled particles expressed av particles per liter. It seems possible that even small variations in the standard breathing maneuver could result in significant changes.

2. The amount of analyzed particles are in exceptionally low range – nanogram. The membrane collecting the particles were divided in 2 by hand after collection. One of the samples were used for SpA/Albumin and one sample was used for MS. Seems highly likely that personal differences and imprecise cutting of the membrane could potentially be a highly likely source of source of error. Was the amount off particles confirmed on the membrane after dividing by hand? Dividing 120ng into exactly 60 ng in each seems highly unlikely.

3. The results are somehow contradictory. In the introduction the authors discuss previous findings. COPD patients has been found to have less surfactant, DPPC, and POPC. However, it seems like the results form the current study implicate the opposite.

4. Age is well known to affect the lung. In the current study the age ranges from 38 - 83. The authors state that has taken the age into consideration in the statistical regression model. Please explain how the age was considered using their pexa method. Known age differences using the pexa method for sampling etc.

5. The pack years of the patients are presented in the supplement and taken into consideration by the authors in their analyses. Patient number 50, 53 and 65 have a reported pack year of 999 years. Seems unlikely and probably an error. However, the analyses seem to be based on those numbers why one should consider a recalculation of the study.

6. The study is based on a rather small cohort, which could be totally fine. However, the group seems to be highly heterogenous. For example, pack year range between the groups: 3-71 pack years in the former smoking group and 1-55 pack years in the current smokers group. Please explain the selection of groups and how these differences might have inflicted the results.

7. Please add information regarding the former smoker group - time from last smoke until the sampling. Could the results also alter if the patient smoked directly before? Do the authors have information about time from last smoke until sampling in the smokers group? This should be added and taken into consideration.

8. Why was the biomarkers SpA, albumin, DPPC, and POPC selected? Looking at the authors publication list and history it seems that the authors have invented the pexa method some years ago and these are the only biomarkers that have ever been able to be analyzed using this method.

9. A main issue regarding the current study is that the biomarkers SpA, albumin, DPPC, and POPC measured in the study is not validated with for example another method. This should be added.

Reviewer #2: Peer review of manuscript ID: PONE-D-21-07178

Current smoking alter phospholipid- and surfactant protein A levels in small airway lining fluid: an explorative study on exhaled breath

COMMENTS TO THE AUTHORS

First of all I’d like to compliment the authors with the well executed study and the excellent way they reported their results. The field of exhaled breath research holds great promise for the future of (clinical) medicine. And as the results of this study have shown, also for basic understanding of human physiology. My compliments and thank you for inviting me to have a say about it.

The manuscript covers all essential criteria: original research is presented; collection of data, statistics and analyses are performed well and described in sufficient detail; conclusions are supported by the data; and the data set has been made available.

Nevertheless I do have a very small amount of issues which – when addressed – may improve the quality of this manuscript to some extent:

1. Subjects were classified as ‘current’, ‘former’ and ‘never’ smokers. Current smokers are defined as ‘having smoked cigarettes on a regular daily basis for at least a year’. I’m wondering: do you have any data on how many cigarettes a day they actually smoked, since 20 cigarettes a day versus 2 cigarettes a day may make a lot of difference within this group? How may this binary approach of current smoking (smoking “yes” or “no”) influence your results? If no data is available, this could represent a limitation of your study and should thus be mentioned.

2. Line 91 states “The present study aimed to explore the long-term effects of tobacco smoking on [….]”. What do you mean by long-term? You did not specify how long patients in the ‘former smoker’ group had quit smoking, again you chose a binary approach: former smoking yes or no, instead of making sub-divisions within the group. Because of that, I think you cannot say anything about long-term effects. Maybe just leave out the word ‘long-term’ in this sentence, or otherwise try to specify.

3. You state that your findings might lead to a useful tool for the identification of patients at risk of developing diseases affecting small airways such as COPD. As a clinician I am very interested in the clinical application and value of a new biomarker and/or test. Therefore I’d like to invite you to take it one step further and elaborate on how this breath test would benefit future early COPD patients..? Could be in one or two sentences. I think you can really point out to your readers why exhaled breath analysis can make a great contribution to future clinical practice.

4. Line 139, last word: ‘was’ should be ‘were’.

5. Line 357, sixth word: ‘needs’ should be ‘need’.

Again, thank you for your excellent work and please continue your research on breath analysis.

6. PLOS authors have the option to publish the peer review history of their article (what does this mean?). If published, this will include your full peer review and any attached files.

Reviewer #1: No

Reviewer #2: **Yes: **Dr PMP van Oort, MD PhD

---

## [Author Response · Author response to Decision Letter 0]

27 May 2021

EDITOR comments and our replies:

Comment 1. Please ensure that your manuscript meets PLOS ONE's style requirements, including those for file naming. The PLOS ONE style templates can be found at

and

Reply: We have tried to follow your style requirements of the main body of the manuscript, likewise file naming have been updated according to PLOS ONE´s style requirements.

Comment 2. In your Methods section, please provide additional information about the participant recruitment method and the demographic details of your participants. Please ensure you have provided sufficient details to replicate the analyses such as: a) the recruitment date range (month and year), b) a description of any inclusion/exclusion criteria that were applied to participant recruitment, d) a description of how participants were recruited, and e) descriptions of where participants were recruited and where the research took place.

Reply: Thank you! We have modified the information about the participant recruitment method and the demographic details of the study participants as suggested under the method section.

Comment 3. Please provide a sample size and power calculation in the Methods, or discuss the reasons for not performing one before study initiation.

Reply: In view of this novel explorative study, where no a priori information on the variation of most PEx-variables and the expected difference between smoking groups were available, a sample size estimation was not possible to perform. Thus, the sample size was based on general idea of what sufficient analytical sample is needed from previous pilots and studies conducted within the team in this research context.

Comment 4. Please provide the product number and any lot numbers of the ELISA kits purchased for your study.

Reply: The product number and lot numbers of the ELISA kits used have been updated in the method section in the manuscript.

Comment 5. Please note that PLOS does not permit references to 'data not shown.' Authors should provide the relevant data within the manuscript, the Supporting Information files, or in a public repository. If the data are not a core part of the research study being presented, we ask that authors remove any references to these data. Please add a citation to support this phrase or upload the data that corresponds with these findings to a stable repository (such as Figshare or Dryad) and provide and URLs, DOIs, or accession numbers that may be used to access these data.

Reply: Data previously referred to as “data not shown” has now been included in the already existing figures.

Comment 6. During your revisions, please confirm whether the wording in the title is correct and update it in the manuscript file and online submission information if needed. Specifically, please consider whether it should read "Current smoking alters..." rather than "alter".

Reply: The title has been updated and now reads “Current smoking alters phospholipid- and surfactant protein A levels in small airway lining fluid: an explorative study on exhaled breath”.

REVIEWER #1 comments and our replies:

Comment 1: Regarding the submitted manuscript Current smoking alter phospholipid- and surfactant protein A levels in small airway lining fluid: an explorative study on exhaled breath-

The authors aimed to explore the effect of smoking on the composition and number of exhaled particles in a smoker-enriched study population.

The study consists of 102 subjects (29 never smokers, 36 former smokers and 37 current smokers) aged 39 to 83 years (median 63). Samples were optioned at one time point for all included patients.

The contents of surfactant protein A and albumin in exhaled particles was quantified with immunoassays and the contents of the phospholipids dipalmitoyl- and palmitoyl-oleoyl- phosphatidylcholine with mass spectrometry.

I have some major concern regarding the current manuscript

Reply: We sincerely thank you for reading and reviewing our manuscript. We have tried to address your major concerns, reflected by your comments below, and hope that this have clarified and improved the manuscript.

Comment 2: Please explain the rational for analyzing number of exhaled particles expressed as particles per liter. It seems possible that even small variations in the standard breathing maneuver could result in significant changes.

Reply: Thank you for this relevant reflection regarding how to address number of exhaled particles (PEx), and likewise the repeatability and reproducibility of using the presented standard breathing maneuver. How to best address number PEx to make it comparable between subjects is still not fully clear. By dividing number PEx with total volume of exhaled air, we try to normalize PEx based on different lung sizes. PEx/exhalation is an alternative way of expressing number PEx, reducing the variation between gender. In present study, we chose to use number PEx/L and adjusted for age and gender in the regression analysis. 

Kokelj and colleagues recently presented data on variability in number of particles, expressed as both per exhalation and per Liter, in healthy subjects [1]. The intra- and inter-individual variation in both these PEx-variables, collected with the standardized breathing maneuver, is only partly explained by age, gender and lung size [2]. We try, within reasonable efforts, to minimize the impact of factors in the breathing maneuver that we think can influence the composition of exhaled particles. For instance, inhalation- and exhalation flow, likewise the breath-hold at residual volume, are therefore carefully monitored by the instructor. 

We have added to the discussion-section, the following limitations (page 18, line 376-380); “The intra- and inter-individual variation in PEx-variables, collected with the standardized breathing maneuver has not been addressed in current study. For practical reasons, it was not possible to perform repeated measures or measures at the same time-point during the day, which might have reduced the bias of diurnal variations in PEx-variables if taken into account [1]. However, the recently published study by Hussain-Alkhateeb et al showed similar results as ours [3].”

Comment 3: The amount of analyzed particles are in exceptionally low range – nanogram. The membrane collecting the particles were divided in 2 by hand after collection. One of the samples were used for SpA/Albumin and one sample was used for MS. Seems highly likely that personal differences and imprecise cutting of the membrane could potentially be a highly likely source of source of error. Was the amount off particles confirmed on the membrane after dividing by hand? Dividing 120ng into exactly 60 ng in each seems highly unlikely.

Reply: The impactor is designed with 10 nozzles through which particles of a certain size passes and impacts on the membrane as 10 spots, see fig 1. These spots are separated with a space in between them, with 5 placed to the left and 5 to the right, making it easy to separate the two halves without being in contact with any of the spots. The nozzles of the impactor was designed with the intention of having the possibility to split the sample in half without touching the sample. The splitting of the sample in two equal parts allow the use of different extraction protocols required for lipid-analysis (organic) and protein-analysis (aqueous). We have extensive data of how the particles deposit on the collection surface based on for example TOF-SIMS analysis (a method where lipids are analyzed directly on the collection substrate without extraction). On the mirror like finish of silicon wafers, it is quite apparent visually (fig.1). Furthermore, during method development for analytical assays, identical samples are needed for evaluating different protocols of extraction and reproducibility, it is imperative to distinguish sampling errors to analytical errors to identify the weakest link. This testing have confirmed that a sample on PTFE substrate split in half will have very similar analyte amounts. This was evaluated for three consecutive collections where substrate was split in half, RSD% for split samples were only 5-10% whereas RSD% for the three samples in total were 10-15%, based on lipids PC16:0/16:0 and PC16:0/18:1 analyzed with LC-MSMS. 

Figure 1. A 25 mm Silcon wafer with sample spots

Comment 4: The results are somehow contradictory. In the introduction the authors discuss previous findings. COPD patients has been found to have less surfactant, DPPC, and POPC. However, it seems like the results form the current study implicate the opposite.

Reply: Thank you for pointing this out, the comparison between current and previous studies seem indeed somewhat contradictory. To make this comparison less inconsistent, we have tried to clarify our hypothesis in the discussion section, as follows (page 14, line 295-303); “It seems likely that smoking may induce changes in the surfactant composition in small airways as a protective mechanism, but at certain stage this will not suffice in susceptible individuals, and pathological processes will take over. By this time the surfactant function will deteriorate, airways start to close, emphysema will develop and the lung function will start to decline. This was indicated by the present results, with a significant association between current smoking and higher DPPC in PEx among subjects with normal lung function as compared to no association between smoking and lipids in PEx in subjects with abnormal lung function (Fig 2D). Previous finding by Laerstad et [4], that subjects with more severe COPD (Gold II-IV) had lower number of exhaled particles, further support this.

Comment 5: Age is well known to affect the lung. In the current study the age ranges from 38 - 83. The authors state that has taken the age into consideration in the statistical regression model. Please explain how the age was considered using their pexa method. Known age differences using the pexa method for sampling etc.

Reply: Age was included as an adjustment factor in the quantile regression, which showed significant contribution to the model. Previous studies have shown age to be a significant predictor of certain PEx-variables [2, 3]. In current study, the number of subjects were too small to draw any further conclusions on how age, in relation to smoking, affects the PEx-variables, wherefore we only chose to adjust for age to minimize the associated heterogeneity in the smoking groups.

Comment 6: The pack years of the patients are presented in the supplement and taken into consideration by the authors in their analyses. Patient number 50, 53 and 65 have a reported pack year of 999 years. Seems unlikely and probably an error. However, the analyses seem to be based on those numbers why one should consider a recalculation of the study.

Reply: 999 indicated missing values and have been treated as such in the analysis. 

Comment 7: The study is based on a rather small cohort, which could be totally fine. However, the group seems to be highly heterogenous. For example, pack year range between the groups: 3-71 pack years in the former smoking group and 1-55 pack years in the current smokers group. Please explain the selection of groups and how these differences might have inflicted the results.

Reply: Thank you for this comment. The recruiting procedure have been clarified and now reads (page 5, line 100-106); 102 subjects, aged 39-83 years, were recruited to participate in an extended study protocol in the follow-up study of the population-based INTERGENE-ADONIX cohort [5]. Recruited subjects had all given their written informed consent in the follow-up study to be contacted for participation in the extended study protocol. Eligible for inclusion were subjects whom had managed to perform spirometry and had reported their smoking status and smoking history in the follow-up study. Exclusion criteria were ongoing respiratory tract infection, myocardial infarction past month, and/or pregnancy in last trimester. All current smokers were invited to participate, whereas former- and never-smokers were invited randomly. 

The selection of groups was mainly done with the intention to see whether an ongoing tobacco exposure may be associated with altered PEx-variables concentrations, in comparison to that of never-smokers. Furthermore, we aimed to explore whether this potential alteration would remain after smoking cessation. It would of course be of interest to further explore how the amount smoked affects the PEx-variables, but for doing that we believe we need to have a larger study population. Anyhow, with age-adjustment in current study, we have minimized the associated heterogeneity in the groups. It is quite likely that higher pack-year range in former smoker is due to that older group present in that group. 

Comment 8: Please add information regarding the former smoker group - time from last smoke until the sampling. Could the results also alter if the patient smoked directly before? Do the authors have information about time from last smoke until sampling in the smokers group? This should be added and taken into consideration.

Reply: Thank you for this interesting reflection regarding the acute and time-course of effect of smoking on PEx-variables. Of course this would be of interest to study further, but that would demand another study-setup which hopefully could be done in a near future. In current study, smokers were “restricted to withdraw from smoking 1 hour prior to the test.” Unfortunately, no more information about last smoke was collected. The definition of former smokers in current study was that they “had smoked on a regular basis, but had stopped smoking for one year or more before the clinical examination.” Due to the fairly low number of former smokers, likewise the wide range of pack-years among smokers (as discussed in reply to comment 7), time since smoking cessation will not be addressed in current study but instead more in depth in an upcoming study with a larger number of former smokers. 

Comment 9: Why was the biomarkers SpA, albumin, DPPC, and POPC selected? Looking at the authors publication list and history it seems that the authors have invented the pexa method some years ago and these are the only biomarkers that have ever been able to be analyzed using this method.

Reply: Findings from earlier study by Laerstad et al [4] indicated that SP-A and albumin in PEx potentially are interesting biomarkers for smoke pathology but further studies including more subjects were asked for, and also comparison of PEx-variables in smokers without COPD. Methods for analyzing surfactant protein A, albumin, DPPC and POPC have successfully been developed and shown high reproducibility at the present small lab. Concerning lipids in the surfactant, there are animal studies (mice) indicating a rather large effect on lipid composition after exposure to cigarette smoke [6]. We therefore chose these biomarkers, and the number of PEx, to further explore their potential as biomarkers for smoke pathology, with special focus on current smokers without COPD. 

Comment 10: A main issue regarding the current study is that the biomarkers SpA, albumin, DPPC, and POPC measured in the study is not validated with for example another method. This should be added.

Reply: Thank you for this comment. The PExA method have previously been compared with broncho-alveolar lavage (BAL) fluid and bronchial wash (BW) in a study of Behndig and colleagues who compared SP-A and albumin in PEx to BAL fluid and to BW [7]. The results showed PEx-content to be similar to BAL but not to BW. We have added this reference to the introduction section (page 4, line 81-82); SP-A levels in PEx have also been shown to be highly correlated with that of BAL fluid [7].

In an early study, SP-A in PExA were compared to that in Exhaled breath condensate (EBC) and in serum [8]. We are currently working on a manuscript comparing lipids in BAL and PExA, taken from the same subject, that will be submitted within short, showing in general extremely good correlation for PC class of lipids. Pearson correlation of log transformed data (for normal distribution of model residuals) ranged between 0.68-0.9, and was recently presented at American Respiratory Society (ATS) [9]. Nevertheless, the PEx-matrix is neither collected in the same way, nor at the same site as for instance BAL or blood, wherefore one would not suspect these different matrixes to reflect the same thing equally. 

REVIEWER #2 comments and our replies:

Comment 1: First of all I’d like to compliment the authors with the well executed study and the excellent way they reported their results. The field of exhaled breath research holds great promise for the future of (clinical) medicine. And as the results of this study have shown, also for basic understanding of human physiology. My compliments and thank you for inviting me to have a say about it. The manuscript covers all essential criteria: original research is presented; collection of data, statistics and analyses are performed well and described in sufficient detail; conclusions are supported by the data; and the data set has been made available. Nevertheless I do have a very small amount of issues which – when addressed – may improve the quality of this manuscript to some extent.

Reply: Thank you so much for this supportive comments and shared enthusiasm about the field of exhaled breath research. We have tried to answer the highly relevant comments regarding some issues that should be addressed and we hope that this has improved the quality of the manuscript. 

Comment 2: Subjects were classified as ‘current’, ‘former’ and ‘never’ smokers. Current smokers are defined as ‘having smoked cigarettes on a regular daily basis for at least a year’. I’m wondering: do you have any data on how many cigarettes a day they actually smoked, since 20 cigarettes a day versus 2 cigarettes a day may make a lot of difference within this group? How may this binary approach of current smoking (smoking “yes” or “no”) influence your results? If no data is available, this could represent a limitation of your study and should thus be mentioned.

Reply: Thank you for this comment on the complexity of how to target the exposure cigarette smoke in this context. The intensity in smoking can of course be analyzed in different ways, as for example in pack years or in cigarettes per day or in years smoked. In current study, the subjects reported the year when starting to smoke and the number of cigarettes smoked per day during the different years when smoking, and from that, we have calculated pack-years. 

In this exploratory set-up, with a limited number of subjects with history of smoking, we mainly used pack years as a variable to describe the smoking load. Neverteless, in a sensitivity study, we tested both pack years and smoking status as variables in the regression analysis and found that being a current smoker affected the PEx-variables stronger than an increase in pack years, even if the latter also was significant, which is why we used smoking status in the presented analysis. In an upcoming study with more subjects, we will address both smoking status and the smoking intensity likewise the duration of smoking more thoroughly.

Comment 3: Line 91 states “The present study aimed to explore the long-term effects of tobacco smoking on [….]”. What do you mean by long-term? You did not specify how long patients in the ‘former smoker’ group had quit smoking, again you chose a binary approach: former smoking yes or no, instead of making sub-divisions within the group. Because of that, I think you cannot say anything about long-term effects. Maybe just leave out the word ‘long-term’ in this sentence, or otherwise try to specify.

Reply: We have, as suggested, leaved out the word “long-term” and rephrased the sentence which now reads (page 5, line 93-95); “The present study aimed to explore potential biomarkers for inflammation in small airway lining fluid; i.e. lipids (DPPC and POPC) and proteins (SP-A and albumin) in PEx and the PEx number concentration, in cigarette smokers.” 

Comment 4: You state that your findings might lead to a useful tool for the identification of patients at risk of developing diseases affecting small airways such as COPD. As a clinician I am very interested in the clinical application and value of a new biomarker and/or test. Therefore I’d like to invite you to take it one step further and elaborate on how this breath test would benefit future early COPD patients..? Could be in one or two sentences. I think you can really point out to your readers why exhaled breath analysis can make a great contribution to future clinical practice.

Reply: Thank you for showing your enthusiasm in the potential of exhaled breath analysis, and for giving us the push forward to take the chance to further elaborate on the potential usefulness of the PExA method and its potential in the field of early COPD. 

In the discussion section, we already wrote (page 18, line 390-392); “In a longer run, it might be a useful tool for the screening of large populations and may facilitate the identification of subjects at risk of developing severe diseases affecting small airways, such as COPD.”

We have now also added the following sentence to the discussion section (page 18, line 388-390); “At individual level, it is easy to neglect that smoking implies an increased risk for severe outcomes, but if there are signs of ongoing inflammation and an improved individual risk-assessment, it may help people to quit smoking.” 

Comment 5: Line 139, last word: ‘was’ should be ‘were’.

Reply: Thank you for pointing this out. This is corrected. 

Comment 6: Line 357, sixth word: ‘needs’ should be ‘need’.

Reply: Thank you for pointing this out. This is corrected.

Comment 7: Again, thank you for your excellent work and please continue your research on breath analysis.

Reply: We are so grateful for this positive feedback! We will of course continue with our research on breath analysis, as there are so many interesting and possible studies yet to be done.

REFERENCES

1. Kokelj S, Kim JL, Andersson M, Runstrom Eden G, Bake B, Olin AC. Intra-individual variation of particles in exhaled air and of the contents of Surfactant protein A and albumin. PLoS One. 2020;15(1):e0227980.

2. Bake B, Ljungström E, Claesson A, Carlsen HK, Holm M, Olin AC. Exhaled Particles After a Standardized Breathing Maneuver. J Aerosol Med Pulm Drug Deliv. 2017;30(4):267-73.

3. Hussain-Alkhateeb L, Bake B, Holm M, Emilsson Ö, Mirgorodskaya E, Olin AC. Novel non-invasive particles in exhaled air method to explore the lining fluid of small airways-a European population-based cohort study. BMJ open respiratory research. 2021;8(1).

4. Larstad M, Almstrand AC, Larsson P, Bake B, Larsson S, Ljungstrom E, et al. Surfactant Protein A in Exhaled Endogenous Particles Is Decreased in Chronic Obstructive Pulmonary Disease (COPD) Patients: A Pilot Study. PLoS One. 2015;10(12):e0144463.

5. Mehlig K, Berg C, Bjorck L, Nyberg F, Olin AC, Rosengren A, et al. Cohort Profile: The INTERGENE Study. International journal of epidemiology. 2017;46(6):1742-3h.

6. Morissette MC, Shen P, Thayaparan D, Stampfli MR. Disruption of pulmonary lipid homeostasis drives cigarette smoke-induced lung inflammation in mice. Eur Respir J. 2015;46(5):1451-60.

7. Behndig AF, Mirgorodskaya E, Blomberg A, Olin AC. Surfactant Protein A in particles in exhaled air (PExA), bronchial lavage and bronchial wash - a methodological comparison. Respir Res. 2019;20(1):214.

8. Larsson P, Mirgorodskaya E, Samuelsson L, Bake B, Almstrand AC, Bredberg A, et al. Surfactant protein A and albumin in particles in exhaled air. Respir Med. 2012;106(2):197-204.

9. Larsson P, Biller H, Koster G, Postle A, Olin A-CC, Hohlfeld JM. Exhaled Breath Particles as a Novel Tool to Study Lung Lipid Composition. C31 COPD BASIC MECHANISMS. p. A4742-A.

---

## [Decision Letter · Decision Letter 1]

14 Jun 2021

Current smoking alters phospholipid- and surfactant protein A levels in small airway lining fluid: An explorative study on exhaled breath

PONE-D-21-07178R1

Dear Dr. Viklund,

We’re pleased to inform you that your manuscript has been judged scientifically suitable for publication and will be formally accepted for publication once it meets all outstanding technical requirements.

Kind regards,

Stelios Loukides

Academic Editor

PLOS ONE

Additional Editor Comments (optional):

Reviewers' comments:

Reviewer's Responses to Questions

**Comments to the Author**

1. If the authors have adequately addressed your comments raised in a previous round of review and you feel that this manuscript is now acceptable for publication, you may indicate that here to bypass the “Comments to the Author” section, enter your conflict of interest statement in the “Confidential to Editor” section, and submit your "Accept" recommendation.

Reviewer #1: All comments have been addressed

Reviewer #2: All comments have been addressed

2. Is the manuscript technically sound, and do the data support the conclusions?

Reviewer #1: Partly

Reviewer #2: Yes

3. Has the statistical analysis been performed appropriately and rigorously? 

Reviewer #1: Yes

Reviewer #2: Yes

4. Have the authors made all data underlying the findings in their manuscript fully available?

Reviewer #1: Yes

Reviewer #2: (No Response)

5. Is the manuscript presented in an intelligible fashion and written in standard English?

Reviewer #1: Yes

Reviewer #2: (No Response)

6. Review Comments to the Author

Reviewer #1: The authors have adressed all my questions and given answers in a point by point letter. The senior authors seem to be the owner of the PExA company, and some of the co authors seems to be stake holders in the same company, therefore it should be clear for the readers that authors do have a conflict of interest.

Reviewer #2: (No Response)

7. PLOS authors have the option to publish the peer review history of their article (what does this mean?). If published, this will include your full peer review and any attached files.

Reviewer #1: No

Reviewer #2: No

---

## [Editor Report · Acceptance letter]

17 Jun 2021

PONE-D-21-07178R1 

Current smoking alters phospholipid- and surfactant protein A levels in small airway lining fluid:  An explorative study on exhaled breath 

Dear Dr. Viklund:

I'm pleased to inform you that your manuscript has been deemed suitable for publication in PLOS ONE. Congratulations! Your manuscript is now with our production department. 

Kind regards, 

on behalf of

Dr. Stelios Loukides 

Academic Editor

PLOS ONE